# Breathless Aftermath: Post-COVID-19 Pulmonary Fibrosis

**DOI:** 10.3390/v17081098

**Published:** 2025-08-09

**Authors:** Dharanya Muthiah, Kishore Vaddadi, Lin Liu

**Affiliations:** 1Oklahoma Center for Respiratory and Infectious Diseases, Oklahoma State University, Stillwater, OK 74074, USA; dharanya.muthiah@okstate.edu (D.M.); kvaddad@okstate.edu (K.V.); 2The Lundberg- Kienlen Lung Diseases and Infection Laboratory, Department of Physiological Sciences, Oklahoma State University, Stillwater, OK 74074, USA

**Keywords:** long COVID, pulmonary fibrosis, biomarker, inflammation

## Abstract

A significant number of individuals recovering from COVID-19 continue to experience persistent symptoms, collectively referred to as Post-Acute Sequelae of SARS-CoV-2 infection (PASC), or long COVID. Among these complications, post-COVID-19 pulmonary fibrosis (PC19-PF) is one of the most severe long-term outcomes, characterized by progressive lung scarring, chronic respiratory impairment, and reduced quality of life. Despite the increasing prevalence of PC19-PF, its underlying mechanisms remain poorly understood. In this review, we provide a comprehensive overview of PC19-PF, including its epidemiology, clinical manifestations, diagnostic strategies, and mechanistic insights. Additionally, we highlight the shared pathways between PC19-PF and other fibrotic lung diseases and discuss emerging therapeutic strategies. This review consolidates emerging insights from both clinical and experimental studies to advance our understanding of PC19-PF pathogenesis and guide the development of mechanism-based therapeutic approaches.

## 1. Introduction

In December 2019, a highly transmissible disease emerged from a novel coronavirus known as SARS-CoV-2. It was officially designated as Coronavirus Disease 2019 (COVID-19) by the World Health Organization (WHO) on 11 February 2020 [1]. This disease quickly became a long-term epidemic with widespread community transmission. Notably, a considerable proportion of cases were asymptomatic or mild, imposing considerable strain on public health worldwide. Consequently, it escalated into a global pandemic in March 2020. Despite widespread vaccination efforts, COVID-19 continues to burden public health.

COVID-19 survivors may experience symptoms that persist well beyond the resolution of the acute illness, a condition referred to as “long COVID” or Post-Acute Sequelae of SARS-CoV-2 infection (PASC). This condition can affect multiple organ systems, including the pulmonary, hematologic, cardiovascular, neuropsychiatric, renal, endocrine, gastrointestinal, hepatobiliary, and dermatologic systems. Common persistent symptoms include dyspnea, cardiac dysfunction, fatigue, arthralgia, cognitive impairments, and evidence of multiorgan damage—particularly chronic lung disease [2]

Among the wide spectrum of long COVID manifestations, one particularly concerning complication is the development of pulmonary fibrosis, known as post-COVID-19 pulmonary fibrosis (PC19-PF). This condition is characterized by progressive scarring and remodeling of the pulmonary parenchyma, leading to chronic respiratory impairment and a decline in quality of life [3,4,5]. Unlike idiopathic pulmonary fibrosis (IPF) or other interstitial lung diseases (ILDs), PC19-PF arises as a sequela of viral pneumonia and often shows a different clinical pattern, including distinct imaging features and, in some cases, partial recovery over time [6]. However, the pathophysiological mechanisms driving PC19-PF remain incompletely understood.

In this review, we provide a comprehensive overview of PC19-PF, focusing on its clinical presentation, diagnostic approaches, underlying mechanisms, and emerging therapeutic and prevention strategies. By synthesizing current evidence from clinical and experimental studies, we aim to enhance the understanding of this debilitating condition and support the development of targeted interventions for its prevention and management.

## 2. Post-COVID-19 Pulmonary Fibrosis: Epidemiology, Risk Factors, and Diagnosis

### 2.1. Epidemiology

Multiple studies have reported persistent pulmonary dysfunction and fibrotic lung changes among COVID-19 survivors. However, the reported prevalence of PC19-PF varies considerably depending on the study population (general COVID-19 population vs. hospitalized/severe cases), geographical setting, and follow-up duration.

A meta-analysis study involving 2018 COVID-19 survivors from various cohort and cross-sectional studies conducted in different countries, China, Egypt, UK, USA, Iran, and Italy, reported an overall prevalence of PC19-PF of 44.9% (907 cases) in patients [7]. This estimate primarily reflects a mixed population of hospitalized and non-hospitalized COVID-19 survivors. Importantly, this meta-analysis excluded studies that focused only on severe or critical COVID-19 cases to avoid skewed results and ensure the findings relevant to a wider group of patients. The inclusion criteria mandated RT-PCR-confirmed COVID-19 recovery (based on either negative follow-up RT-PCR, discharge from hospital, or being asymptomatic for at least one month), as well as a clearly defined fibrotic outcome that distinguished fibrotic from non-fibrotic cases. Despite this, the meta-analysis was limited by substantial heterogeneity in assessment time points, study scale, sample sizes, and definitions of fibrosis, which could influence prevalence estimates [7].

Similarly, another meta-analysis comprising nine studies (eight cohort studies and one cross-sectional study) conducted in China, Egypt, and Pakistan included a total sample of 1241 post-COVID-19 patients and revealed a pooled prevalence of 54.04% based on a random-effects model [8]. These studies also primarily included hospitalized individuals, and the inclusion criteria ranged from mild to severe cases, though detailed stratification was not uniform across all datasets. Variability in prevalence likely stems from differences in study design, patient populations (e.g., inclusion of ICU or non-ICU patients), outcome definitions, and post-COVID assessment periods ranging from weeks to several months.

These estimates are comparable to those observed for SARS-CoV (62%)- and MERS-CoV (33%)-associated pulmonary fibrosis [9,10]. The burden of PC19-PF remains notably high, and warrants continued clinical attention.

However, it is essential to interpret these estimates in light of the population studied. The high prevalence rates reported in these meta-analyses largely reflect moderate-to-severe or hospitalized patients and do not represent the general population of COVID-19 survivors. Furthermore, not all individuals with severe or critical COVID-19 go on to develop fibrosis. It has been estimated that only 9% of such patients will develop fibrotic-like changes [11,12], and among these patients, up to 42% may still have signs of lung damage on scans and symptoms of pulmonary fibrosis even two years after their initial COVID-19 infection [13].

Taken together, while the burden of PC19-PF is particularly high among patients with moderate-to-severe or hospitalized COVID-19, its prevalence in the broader population of COVID-19 survivors remains significantly lower. These distinctions are crucial for understanding the true epidemiological impact of PC19-PF and for identifying high-risk subgroups who may benefit from long-term pulmonary monitoring and intervention.

### 2.2. Risk Factors

Several patient- and disease-related factors are associated with an increased risk of developing PC19-PF (Figure 1). Among these, disease severity stands out as the most significant determinant. The extent of lung injury during the acute phase of COVID-19 has a direct correlation with fibrotic outcomes in the post-acute phase. A prospective cohort study during the first wave reported fibrotic changes in approximately 79% of patients with moderate illness and up to 100% of those with severe or critical disease [6]. This suggests that the intensity of lung inflammation and damage during acute illness plays a major role in driving long-term fibrotic remodeling.

In addition to severity, mechanical ventilation and ICU admission are strongly linked to PC19-PF [6,14]. According to a prospective study conducted during the early pandemic by McGroder et al. [15], fibrotic abnormalities were seen in 72% of patients who underwent invasive mechanical ventilation, compared to only 20% of those who did not. This could be due to ventilator-induced lung injury, which results from prolonged mechanical ventilation and variations in pressure or volume settings. These factors can trigger the release of pro-inflammatory mediators, worsening lung injury and increasing the risk of long-term fibrosis in survivors [16].

The SARS-CoV-2 variant type may also influence the extent of lung damage and fibrosis risk. A retrospective observational study conducted at the Cleveland Clinic (March 2021–March 2022) analyzed 2779 SARS-CoV-2-positive patients confirmed by next-generation sequencing of nasopharyngeal samples. The study compared patients infected with the Alpha (*n* = 1153), Gamma (*n* = 122), Delta (*n* = 808), and Omicron (*n* = 696) variants. Alpha, Gamma, and Delta infections were associated with more severe disease outcomes: ~12% hospitalization, ~3% ICU admission, and 6–9% oxygen requirement. In contrast, Omicron caused a substantially milder disease, with hospitalization of 5.9% of patients, ICU admission of 1%, and oxygen use by 3.4%. These differences are likely due to the reduced replication of Omicron in the lower respiratory tract and weaker suppression of interferon responses. Despite Omicron having the highest rate of breakthrough infections (73.7%), severe outcomes were largely driven by host-related factors such as older age and comorbidities. This underscores that while viral virulence contributes to the fibrosis risk, patient susceptibility remains critical [17]. Interestingly, a study showed that individuals infected with the Delta variant were found to induce more severe and extensive fibrosis-like changes compared to those infected with earlier strains, suggesting that different variants may influence the extent of long-term lung damage [18].

Age is another important risk factor. A follow-up study of 287 COVID-19 patients during the first wave (followed between 90 and 150 days after disease onset) reported that older age and a higher body mass index (BMI) were significantly associated with persistent fibrotic changes [19]. Age-related immunosenescence, increased oxidative stress, the accumulation of senescent cells, and extracellular matrix dysregulation impair the lungs’ capacity for viral clearance and repair, predisposing patients to fibrosis [20,21,22]. In support of this, a study conducted in COVID patients between 1 March and 15 May 2020 found that for every 10% reduction in age-adjusted telomere length, the risk of developing post-COVID fibrosis increased by 1.35 times [15].

However, not all studies find a consistent correlation between age or BMI and fibrosis. For example, Aul et al. [23] found no significant difference in age or BMI between patients with and without fibrotic changes in a cohort of COVID-19 patients infected during the first wave, highlighting that additional factors modulate the fibrosis risk.

Sex differences also influence outcomes. The development of PC19-PF appears to be more prevalent in men. In one study, men were nearly three times more likely to develop PC19-PF [23]. Similarly, a retrospective study in Italy involving 90 patients with COVID-19 pneumonitis during the first wave reported that 25.5% of them developed fibrosis, with men comprising 65% of these patients [24]. The heightened risk of pulmonary fibrosis in men may be attributed to their relatively weaker immune response to SARS-CoV-2, which often results in more severe disease manifestations, such as acute respiratory distress syndrome (ARDS) and the need for mechanical ventilation. These complications are strongly associated with extensive lung injury and are well-recognized precursors to fibrotic remodeling [25,26,27]. However, some studies have found no significant sex-based correlation with PC19-PF, indicating the need for further investigation [19,28].

Comorbidity increases the risk of developing fibrosis and has a detrimental effect on PC19-PF patient survival rates [29]. A retrospective study found that 23 out of 90 patients with COVID-19 pneumonitis suffered from pulmonary fibrosis (25.5%), and many of them had comorbidities (78.2%), including hypertension (47.2%), diabetes (34.7%), and liver disease (1.1%) [24]. PC19-PF is nearly three times more prevalent in COPD patients [7]. Another study showed that 90 out of 173 COVID-19 survivors had PC-19 PF, and cardiovascular diseases were the most prevalent comorbidity seen in the patients (*n* = 72, 41.6%) [30]. These diseases promote a chronic inflammatory state and impair tissue repair mechanisms, compounding the effects of virus-induced lung injury.

Furthermore, several lifestyle choices, like smoking and heavy drinking, can cause chronic epithelial injury, oxidative stress, and inflammation that could accelerate the fibrotic process [31,32]. Additionally, use of drugs such as rituximab, chemotherapy agents, and corticosteroids can significantly increase the risk of developing lung fibrosis following COVID-19 infection. These drugs could suppress immune function, leading to more severe infections and higher chances of progressing to fibrosis [33].

Taken together, these findings demonstrate that while many factors contribute to the development of PC19-PF, the severity of the acute illness remains the strongest predictor. Other factors, such as age, sex, comorbidities, viral variant, lifestyle, and drug exposure, act synergistically to modulate individual susceptibility.

### 2.3. Diagnosis

#### 2.3.1. Diagnostic Criteria and Clinical Features

The aftermath of a COVID-19 infection imposes a considerable burden on the respiratory system, particularly the lungs [34,35]. PC19-PF is typically diagnosed based on clinical, functional, and radiological criteria. According to current definitions, it requires at least one of the following to be present at the three-month follow up: persistent respiratory symptoms (e.g., dyspnea and dry cough); unexplained resting or exertional hypoxemia; a restrictive ventilatory defect (FVC and/or DLCO < 80% predicted); or radiologic abnormalities such as GGOs, fibrotic or subpleural bands, and consolidations [11,36]. The most common symptoms observed in PC19-PF patients at the six-month follow-up after the acute disease were dyspnea (98%), dry cough (91%), fever (70%), dry pain (19%), and chest pain (16%) [37]. Also, 4% to 12% of PC19-PF patients have shown the presence of pathological auscultation sounds, represented by Velcro crackles and wheezing [38].

#### 2.3.2. Pulmonary Function Test

The comprehensive management of PC19-PF requires a meticulous assessment, with emphasis on pulmonary function. In alignment with this, the British Thoracic Society underscores the importance of pulmonary function tests (PFTs) at the three-month post-discharge follow up for all critical COVID-19 survivors. Furthermore, individuals with a mild to moderate disease exhibiting abnormal radiographic findings are also recommended for a PFT [39].

A four-month follow-up study of COVID-19 survivors revealed significant impairments in lung function. The most prominent changes included a reduced diffusing capacity of the lungs for carbon monoxide (DLCO), followed by a decreased forced vital capacity (FVC), reduced alveolar volume (AV), an abnormal forced expiratory volume in one second to forced vital capacity ratio (FEV1/FVC), and impaired exercise capacity, as demonstrated by reduced performance in the Six-Minute Walk Test. These impairments often correlated with the severity of the initial COVID-19 illness [40].

In a single-center prospective cohort study (*n* = 46) conducted between May 2020 and April 2022 and involving mainly Delta variant cases, patients with persistent respiratory symptoms or abnormal radiographs were assessed at 6, 9, and 12 months post-infection. The study found that patients who experienced severe illness exhibited significantly higher rates of DLCO impairment (59% versus 17% of patients with mild/moderate cases) and reduced physical health scores. Abnormalities in PFTs were most pronounced at the 6-month follow-up and were associated with older age, elevated inflammatory markers, and more extensive chest radiograph abnormalities. These findings support the recommendation to perform an initial PFT at 6 months post-COVID-19, with subsequent follow-up testing reserved for patients demonstrating abnormal results [41].

Another prospective study from Israel, conducted during the second wave and involving 168 adults with PCR-confirmed COVID-19 pneumonia, evaluated patients at 3 and 6 months post-infection. Despite initial impairments in DLCO and FVC, no meaningful improvement was noted over time. Interestingly, radiological abnormalities improved, but these did not correlate with pulmonary function. Occupational exposure was a key predictor of a persistent DLCO reduction, supporting the need for individualized follow-up based on risk factors [42]

In Wuhan, China, two prospective studies captured the long-term trajectory of COVID-19 pulmonary outcomes. One longitudinal cohort of 83 patients (median age of 60 years) hospitalized for severe COVID-19 during the first wave (Feb–Mar 2020) and followed for 12 months showed gradual improvement in DLCO and 6MWT, but 33% had a persistent DLCO reduction at one year. HRCT revealed residual GGOs in 24%, but no progressive fibrotic changes [43].

Considering these findings, Mylvaganam et al. [5] recommend regular pulmonary function testing for all patients recovering from acute COVID-19, particularly those with long COVID. These tests enable the early detection of pulmonary abnormalities and establish a baseline for ongoing monitoring. Early identification allows for timely intervention, ultimately enhancing patient outcomes. 

#### 2.3.3. Imaging

During the pandemic, researchers have prioritized investigating radiographic features to comprehend acute COVID-19 infection. In a study conducted by Huang et al. [29], which included a large group of patients from Wuhan, China, more than half of those who underwent CT scans six months after being admitted to the hospital exhibited abnormal radiographic findings. The most common abnormalities included changes in the lungs, such as ground-glass opacities (GGOs) and irregular lines [44]. Furthermore, research revealed that 38% of patients had abnormal chest radiographs around 54 days after being discharged from the hospital, and 9% experienced worsening abnormalities during the study’s follow-up period [45]. These findings underscore the importance of continued investigation into the long-term radiographic manifestations of COVID-19.

Patients with PC19-PF had GGOs (95%), honeycombing (25%), and pulmonary consolidations (9%), according to their CT scans. The fibrotic alterations were shown to be primarily bilateral, peripheral, and in the lower lobes [37,46]. Another study revealed that the most common characteristic findings of PC19-PF on a CT scan or high-resolution computed tomography (HRCT) scan was bilateral GGOs, which may disappear with time (referred as “snowstorm”) or might develop into fibrosis (referred as “honeycomb”) followed by traction bronchiectasis, reticulation, and the presence of fibrotic bands in the parenchyma [11,47]. However, these findings were not typical, and they can vary from patient to patient [48]. Additionally, different SARS-CoV-2 variants can cause variations in imaging presentations. For instance, patients infected with the Omicron variant show a marked decrease in pneumonia incidence, likely due to its reduced tropism for lung tissue, which limits viral replication in alveolar epithelial cells. Consequently, radiographic abnormalities might be less frequent in patients with Omicron infections [49].

Consequently, a comprehensive screening approach involving both pulmonary function tests and cross-sectional imaging has been advocated for individuals presenting with chronic dyspnea post-acute COVID-19 [5]. Furthermore, there is a need for additional studies to investigate the histological characteristics obtained from patients diagnosed with PC19-PF. This is essential to enhance our understanding of this condition.

#### 2.3.4. Biomarkers

There has been minimal investigation regarding the relationship between circulating biomarkers and PC19-PF that could offer valuable insights into the physiological status of the patient or the severity of their illness. Based on the current literature, these biomarkers can be broadly classified into acute phase proteins or reactants, biochemical markers, cytokines and chemokines, mitochondrial regulatory proteins, vascular injury markers, and others (Table 1).

##### Acute Phase Proteins or Reactants

SARS-CoV-2 infection triggers a systemic inflammatory response that elevates acute-phase proteins. C-reactive protein (CRP) is one of the most consistently elevated markers during acute COVID-19 and remains elevated in patients with post-COVID complications, correlating with ongoing systemic inflammation and the fibrosis risk. A study during the first wave assessed hospitalized COVID-19 patients within one year of discharge and reported significantly higher CRP levels during follow-up in those who developed fibrotic lung changes as compared to non-fibrotic individuals. In this study, an elevated erythrocyte sedimentation rate (ESR) and low albumin levels were also linked to PC19-PF, indicating both systemic inflammation and possible nutritional deficiency [51]. Another study reported that elevated levels of CRP and ESR at the time of hospital admission (i.e., during the acute phase) were associated with fibrotic changes observed at 10–12 weeks of follow-up, suggesting that the early inflammatory burden may influence long-term pulmonary outcomes [52]. However, a study by Colarusso et al. found that CRP and C5a levels measured during follow-up (1–3 months post-viral clearance) did not correlate with fibrotic changes, highlighting potential temporal or cohort-specific variability [50].

Additionally, activation of the complement system, particularly C5b-9 (terminal complement complex), has been observed more frequently in post-COVID patients than in vaccinated or healthy controls, supporting the hypothesis of persistent complement-mediated immune injury, though a direct, independent association with fibrosis has not been consistently demonstrated [50].

##### Biochemical Markers

Lactate dehydrogenase (LDH), a well-established marker of tissue injury, is frequently elevated in both the acute and post-acute phases in COVID-19 patients, reflecting underlying lung damage and inflammation [13,53,62]. Similarly, elevated levels of aspartate aminotransferase (AST) and alanine aminotransferase (ALT), typically associated with hepatic injury, have also been observed in patients with severe COVID-19 cases. These elevations are thought to result from pneumonia-induced hypoxia and systemic inflammation, contributing to multi-organ stress and an increased risk of PC19-PF [54].

A comprehensive longitudinal study conducted from March 2020 to December 2023 stratified hospitalized COVID-19 patients into three cohorts based on the wave of infection: Group 1 (first wave), Group 2 (second wave), and Group 3 (third wave). These patients were evaluated at three time points: upon hospital admission, at 3 months, and again at 2 years post-infection. The study demonstrated that elevated LDH levels were strongly associated with the extent of lung injury and the intensity of the inflammatory response, particularly with fibroblast proliferation, a key driver of fibrotic remodeling. Similarly, higher levels of AST and ALT were observed in patients with more severe pulmonary disease, suggesting a hyperinflammatory state that predisposes patients to ARDS and later fibrotic progression. Although LDH and AST levels differed among the groups, Group 2 patients showed comparatively lower levels, indicating the lower severity of the second wave. Both markers remained consistently linked to a heightened risk of developing post-COVID pulmonary fibrosis across all cohorts. LDH emerged as a robust and persistent indicator of fibrotic outcomes [13].

##### Cytokines and Chemokines

In COVID-19, the cytokine storm plays a central role in the development of ARDS and lung fibrosis [28,63]. A study by Hu et al., based on follow-up data from post-COVID patients, showed that reduced IFN-γ expression during convalescence was associated with increased fibrotic susceptibility [28].

Another study conducted on patients infected between July 2020 and April 2021 showed that elevated levels of interleukin-6 (IL-6), IL-1α, and tumor necrosis factor-α (TNF-α) were associated with increased disease severity and fibrotic outcomes during the follow-up study [28,50,55,64]. Notably, higher IL-1α levels, measured during follow-up, were predictive of a nearly threefold increased relative risk of developing fibrosis-like lung changes in post-COVID patients [50]. In contrast, one study found that IL-6 and TNF-α levels, when measured at follow-up, were not significantly different between the fibrotic and non-fibrotic groups [28].

Studies conducted on post-COVID patients during follow-up found that the levels of interferon-alpha (IFN-α) and IFN-β were significantly lower in individuals who developed fibrotic lung changes compared to those without fibrosis. This suggests a protective role for interferons in regulating post-viral immune responses and preventing fibrotic remodeling in the lungs [50,56].

Additionally, the levels of chemokines such as C-X-C motif chemokine ligand 10 (CXCL10) and chemokine (C-C motif) ligand 18 (CCL18) were elevated in the plasma and serum of post-COVID patients with pulmonary fibrosis, suggesting their involvement in chronic inflammation and fibrotic remodeling [50,56].

A major profibrotic cytokine, transforming growth factor-beta (TGF-β), promotes fibroblast activation and extracellular matrix (ECM) deposition, contributing significantly to fibrotic remodeling in post-COVID lungs. Elevated TGF-β levels have been reported in patients with PC19-PF and represent a potential therapeutic target [50,65].

##### Vascular Injury Markers

The cytokine storm triggered by SARS-CoV-2 can damage both lung epithelial and microvascular endothelial cells, leading to elevated levels of vascular injury markers such as vascular cell adhesion molecule-1 (VCAM-1), intercellular adhesion molecule-1 (ICAM-1), and placental growth factor (PIGF). During a 3-month follow-up of patients infected during the first wave, elevated levels of vascular injury markers such as VCAM-1, ICAM-1, and PlGF were detected in post-COVID individuals. Among these, VCAM-1 and ICAM-1 showed positive correlations with residual clinical abnormalities, including persistent computed tomographic (CT) findings and impaired pulmonary function. Specifically, both VCAM-1 and ICAM-1 were significantly negatively correlated with the DLCO predicted percentage, suggesting that endothelial activation and injury may impair alveolar–capillary gas exchange in recovered patients [58].

Conversely, the level of fibroblast growth factor-2 (FGF-2), a cytokine involved in vascular repair, was found to be low during the early phase of infection and tended to increase only later particularly in patients with prolonged illness or more severe lung damage. This pattern, along with delayed elevations in FGF-2 levels, has been observed in individuals who develop post-COVID-19 pulmonary fibrosis [57].

##### Mitochondrial Regulatory Proteins

COVID-19 is associated with mitochondrial impairment and oxidative stress. Elevated levels of mitochondrial biomarkers in post-COVID patients with long-term pulmonary complications suggest a role in PC19-PF pathogenesis. Dysregulation of key mitochondrial regulators such as PTEN-induced kinase 1 (PINK1), dynamin-1-like protein (DNM1L), and mitofusin-2 (MFN2) indicates that SARS-CoV-2 may disrupt mitochondrial homeostasis, contributing to chronic inflammation and oxidative injury in the lungs [56].

##### Other Biomarkers

Another potential marker of pulmonary epithelial cell injury is Krebs von den Lungen 6 (KL-6), a mucin-like glycoprotein primarily found on the surface of type II alveolar epithelial cells and respiratory bronchiolar epithelial cells. Studies have reported that higher levels of KL-6 are associated with interstitial lung disease and severe idiopathic pulmonary fibrosis (IPF) [66,67]. A follow-up study of post-COVID patients revealed that an elevated level of KL-6 can be a predictor of disease progression [68].

Persistent alveolar injury in PC19-PF patients leads to fibroblast activation and extracellular matrix deposition. Consequently, the levels of fibrotic markers such as alpha-smooth muscle actin (α-SMA), collagen type 1 alpha 2 (COL1A2), collagen type 3 alpha (COL3A1), and matrix metalloproteinases 1 and 7 (MMP1, and MMP7) were observed to be elevated in follow-up studies of post-COVID patients with fibrotic lung changes [59,60]. Another study has shown that serum biomarkers of pulmonary fibrosis, including type IV collagen (COL4), laminin (LN), hyaluronic acid (HA), and procollagen III N-terminal peptide (PIIINP), were elevated in PC19-PF patients and may serve as indicators of PC19-PF severity [69].

Another potential biomarker is the level of SARS-CoV-2 antibodies, which may correlate with persistent lung abnormalities. In a prospective cohort of COVID-19 patients discharged between April 2020 and January 2021, evaluations at 6 and 18 months post-discharge revealed that approximately 31% had persistent CT abnormalities. Although lesion volumes decreased over time, those with CT abnormalities showed impaired pulmonary function, including reduced FEV_1_, FEV_1_/FVC, and FVC. Importantly, SARS-CoV-2 IgG levels remained significantly elevated in these patients and positively correlated with the lesion volume. These results suggest that antibody levels could serve as a biomarker reflecting the complex interplay of the immune response, inflammation, and fibrotic remodeling in long-term COVID-19 lung sequelae [61].

## 3. Pathophysiological Mechanisms of Post-COVID-19 Pulmonary Fibrosis

PC19-PF arises from a combination of virus-induced lung injury and impaired tissue repair. The condition primarily progresses through two interconnected mechanisms: injury and aberrant repair of alveolar epithelial cells, particularly type II alveolar cells (AT2), which are crucial for regenerating the lung surface and chronic immune dysregulation, where prolonged inflammation disrupts normal repair and promotes fibrotic remodeling. These processes are further amplified by oxidative stress, prolonged immune cell activation, and elevated levels of pro-fibrotic mediators such as TGF-β. Together, they trigger fibroblast activation and excessive extracellular matrix deposition, leading to irreversible changes in the lung architecture (Figure 2). To better understand the underlying mechanisms and identify potential therapeutic strategies, it is essential to integrate evidence from clinical studies in human patients as well as experimental insights from animal models.

### 3.1. Human Studies

Although multiple interrelated mechanisms contribute to the development of PC19-PF, for clarity, these can be broadly categorized into two main processes: (1) alveolar type II (AT2) cell injury and aberrant regeneration, and (2) dysregulated inflammatory responses, both of which contribute to fibrotic remodeling and impaired lung function.

#### 3.1.1. Alveolar Type II Cell Injury and Aberrant Regeneration

SARS-CoV-2 primarily infects AT2 cells via ACE2 receptors, leading to cellular injury or reprograming that disrupts epithelial repair and induces the epithelial-to-mesenchymal transition (EMT) [70], both of which are known contributing factors to lung fibrogenesis.

##### Dysregulated Regeneration with Transitional Epithelial Cell Accumulation

During normal repair, AT2 cells differentiate into alveolar type I (AT1) cells via an intermediate “transitional” state. However, chronic injury or inflammation can cause the accumulation of these transitional cells, disrupting regeneration and promoting fibrosis [71,72,73].

In a mouse model, IL-1β-primed AT2 cells were shown to give rise to a damage-associated transient progenitor (DATP) state via a HIF1α-mediated glycolytic switch. Although DATPs initially support alveolar regeneration, their pathological accumulation during sustained inflammation impairs AT1 cell differentiation. These observations, supported by single-cell RNA sequencing (scRNA-seq), lineage tracing, and validation in human IPF tissue, highlight inflammation as a key driver of dysregulated epithelial regeneration [71].

A scRNA-seq study of bleomycin-induced lung fibrosis, validated in human fibrotic lung tissues, identified Krt8+ alveolar differentiation intermediates (ADIs) with a squamous morphology, senescence markers, and pro-fibrotic transcriptional profiles (*Itgb6*, *Ctgf*, and *Edn1*). These transitional cells formed communication hubs with macrophages and fibroblasts and persisted in multiple injury models, implicating them in chronic fibrosis [72].

Another mechanistic study using mouse models, organoids, and IPF patient lungs discovered a novel TP53-dependent pre-alveolar type-1 transitional cell state (PATS) during AT2-to-AT1 cell differentiation. PATS cells showed transient senescence and DNA damage during normal repair but persisted abnormally in fibrosis, contributing to impaired regeneration [73].

Recent scRNA-seq analyses of lung tissues from deceased individuals with severe SARS-CoV-2 infection reveal parallel mechanisms. A multi-organ atlas from 32 donors showed a loss of AT2 cells, expansion of fibroblasts and TP63^+^ progenitors, and enrichment of the PATS, indicating failed regeneration. Viral RNA was mainly found in myeloid and endothelial cells, and genome-wide association study (GWAS) integration highlighted AT2 cells as major contributors to disease severity [74]. A second lung-focused study (~116,000 nuclei) identified IL-1β-driven inflammation, impaired T cell responses, and AT2 cell arrest in a DATP state, alongside the expansion of CTHRC1^+^ fibroblasts and fibrosis. Both studies emphasize immune–epithelial dysregulation and regenerative failure as key drivers of COVID-19 lung pathology [75]. Together, these findings suggest that persistent transitional cell states, driven by unresolved inflammation, are a shared pathological feature of both idiopathic and virus-induced lung fibrosis, including PC19-PF.

##### EMT Induction

In a postmortem analysis of lung tissues from 33 individuals who died from COVID-19, fibrosing diffuse alveolar damage (DAD) was observed in those with prolonged illness (>2 weeks), with fibrosis severity increasing over time. Molecular profiling of these lungs showed the marked upregulation of extracellular matrix genes (*COL1A1*, *COL3A1*, *SPP1*, and *LOX*) and fibroblast-associated markers (*CTHRC1* and *SPARC*), indicating active fibrogenesis. Despite the elevated TGF-β1 levels, canonical downstream signaling was suppressed, and immunohistochemistry did not support a major role for the EMT. Notably, viral RNAs, including subgenomic fragments, were detected in lung tissue up to 104 days post-infection, suggesting that persistent viral presence may contribute to ongoing epithelial injury and fibrotic remodeling. Collectively, these findings implicate chronic epithelial damage, aberrant fibroblast activation, and disrupted repair signaling rather than the EMT as central mechanisms in the development of PC19-PF [76].

Conversely, a separate study analyzing bronchoalveolar lavage (BAL) fluid and autopsy tissues from 33 COVID-19 patients identified neutrophil extracellular traps (NETs) as potent inducers of the EMT. Immunohistochemical staining of autopsy tissues confirmed CK7/α-SMA co-expression in alveolar cells, consistent with a partial EMT. In vitro models demonstrated that NETs, especially when combined with macrophage-derived cytokines (TGF-β, IL-8, and IL-1β), synergistically induced an EMT phenotype in alveolar epithelial cells. However, the predominant pathogenic mechanism was NET-mediated fibrogenic reprogramming, suggesting that the EMT plays a context-specific, supportive role rather than serving as the principal driver of fibrosis in COVID-19 patients [77].

##### Oxidative Stress and Receptor Dysregulation

As SARS-CoV2 interacts with ACE2 receptors on alveolar epithelial type II cells in the host, it causes endocytosis and downregulation of the receptor expression, leading to a shift in the renin–angiotensin axis toward pro-fibrotic signaling. A prospective cohort study evaluating BAL fluid and plasma from 65 COVID-19 ARDS patients and 63 ventilated controls found elevated levels of TGF-β1 and N-terminal pro-peptide of type III procollagen (NT-PCP-III), alongside increased levels of oxidative stress markers (malondialdehyde [MDA] and oxidized glutathione [GSSG]) and reduced levels of antioxidants (superoxide dismutase [SOD] and reduced glutathione [GSH]). These alterations correlated strongly with early fibrosis scores on CT and predicted persistent radiological abnormalities in 57% of survivors at the 3-month follow-up, linking a redox imbalance to fibrogenesis [78].

#### 3.1.2. Dysregulated Inflammatory Responses

Persistent immune dysregulation after SARS-CoV-2 clearance is now recognized as a major cause of PC19-PF. Aberrant T cell, macrophage, and neutrophil activation contribute to long-term tissue remodeling and fibrosis progression. Evidence from human cohorts consistently correlates these immunological changes to decreased pulmonary function, radiologic abnormalities, and increased fibrotic gene expression.

##### Tissue-Resident Memory T Cells and Pathogenic CD8+ Subsets

A retrospective observational study investigating immune signatures in aged COVID-19 convalescents (*n* = 10, >60 years) compared to age-matched healthy controls (*n* = 5) analyzed BAL fluid and peripheral blood to assess immune profiles. Despite systemic immune recovery, convalescents exhibited persistent respiratory abnormalities, including CT evidence of ground-glass opacities and reduced pulmonary function (FEV1, FVC, and DLCO). Notably, the BAL fluid of COVID-19 convalescents showed an increased abundance of CD8+ T cells, γδ T cells, and B cells, with SARS-CoV-2-specific memory B and T cells enriched at the site of infection. Respiratory CD8+ T cells, especially tissue-resident memory (TRM)-like subsets, CD69 + CD103, were negatively correlated with lung function and positively with radiologic abnormalities. scRNA-seq further identified pathogenic CD8+ T cell subsets, including cytotoxic cluster 1 (CD103^−/low^ TRM) and CXCR6^hi^ cluster 2 cells, associated with tissue-damaging gene signatures. Complementary murine experiments demonstrated that the depletion of CD8+ T cells after viral pneumonia improved lung function, supporting their pathological role. Importantly, SARS-CoV-2 RNA was undetectable in convalescent BAL fluid, suggesting that immune dysregulation, rather than viral persistence, drives post-COVID lung injury. These findings underscore the contribution of sustained, dysregulated CD8+ T cell responses to a chronic pulmonary impairment following severe respiratory viral infections and highlight potential immune targets for intervention [79].

##### Neutrophil-Driven Inflammation and NETosis

The PROSAIC-19 (prospectively recruited individuals with PCR-confirmed COVID-19 between 1 March 2020, and 1 November 2021) cohort study employed a multiomic analysis of 81 participants, including 46 with severe cases and 18 with mild COVID-19 cases and 17 healthy controls, assessed 3–6 months post-infection. Using blood plasma, PBMCs, nasal swabs, and chest CT, the study demonstrated that over 50% of severe COVID-19 patients had persistent interstitial lung changes, reduced FVC and DLCO, increased total neutrophil counts, elevated plasma myeloperoxidase concentrations (a neutrophil protease), and higher levels of H3R8 citrullinated nucleosomes (a marker of NET formation). Plasma proteomics revealed sustained neutrophil-associated inflammatory signatures, while nasal transcriptomics showed the upregulation of genes such as *CXCL8*, *NLRP3*, and *OASL*. Deconvolution analysis confirmed neutrophil infiltration, and phosphoproteomic profiling of PBMCs indicated the activation of kinases related to NET formation (CDK2, MEK1, and JNK2). In vitro, purified NETs, but not SARS-CoV-2 alone, induced fibrogenic gene expression (*FN1*, *ACTA2*, and *VEGF*) in alveolar epithelial cells, linking neutrophil dysregulation and NETosis to epithelial remodeling. Elevated levels of IL-17C and NET biomarkers were independently associated with fibrotic CT abnormalities, with only partial resolution by 12 months [80].

##### Alveolar Macrophage Depletion and Increased Monocyte-Derived Macrophages

An integrative single-cell transcriptomic study compared severe COVID-19, IPF, and healthy human lungs, incorporating bulk RNA-seq from a TGF-β1-induced rat fibrosis model treated with antifibrotic drugs. COVID-19 and IPF lungs shared key features: the depletion of alveolar macrophages and AT2 cells, increased monocyte-derived macrophages (Mo-Macs) and CD8+ T cells, and upregulation of fibrosis- and inflammation-related genes (e.g., *SPP1*, *S100A8/A9*, and *COL1A1/3*). Macrophages showed metabolic reprogramming with enhanced glycolysis in Mo-Macs and lipid metabolism in alveolar macrophages and enolase 1 (ENO1) gene and phospholipase A2, group VII (PLA2G7) emerging as potential therapeutic targets. Antifibrotics (pirfenidone, nintedanib, and sorafenib) reversed fibrosis-related gene expression in rats, though they did not alter glycolysis or lysosomal protease activity, suggesting the need for adjunct therapies. These findings support shared fibrotic pathways in post-COVID fibrosis and IPF, while identifying novel targets like ENO1 and Granulin (GRN) for PC-19 PF [81].

##### Monocyte Dysfunction

A high-dimensional immunophenotyping study by Bingham et al. compared PBMC samples from sixteen PC19-PF patients (subgrouped into seven early-resolving and nine late-resolving cases), eight IPF patients, and three non-diseased controls, analyzing 71,574 single cells via scRNA-seq and multiplex immunostaining. Late-resolving PC19-PF was marked by monocyte depletion (notably CD14+ and intermediate subsets), downregulation of MHC-II genes (*HLA-DRA* and *HLA-DRB1*), and elevated alarmins (*S100A8*, *S100A9*, and *S100A12*), indicating defective antigen presentation and a pro-inflammatory state. Meanwhile, CD8+ and CD4+ T cells in PC19-PF showed sustained activation (*CD38*, *HLA-DR*, *PD1*, *ICOS*, and *CD69*), enriched in IFN-γ, cytokine signaling, and allograft rejection pathways. In contrast, IPF T cells were characterized by exhaustion and senescence. These findings demonstrate that PC19-PF involves sustained adaptive immune activation, distinct from the aging-associated immunopathology seen in IPF [82].

##### Inflammatory Cytokine Profiles and the Fibrosis Risk

An observational cohort study by Colarusso et al. evaluated 96 participants, including 52 post-COVID patients, 17 healthy controls, and 27 vaccinated individuals, using plasma cytokine profiling, CT imaging, and spirometry at 1–3 months post-infection. Patients who developed fibrotic-like lung changes exhibited significantly lower levels of IFN-β and elevated levels of IL-1α and TGF-β, cytokines associated with chronic inflammation and fibrotic remodeling. CRP, LDH, and terminal complement complex C5b-9 levels were also elevated. Furthermore, over 80% of patients with severe cases developed fibrotic CT features. A cytokine signature defined by low IFN-β levels and high IL-1α and TGF-β levels was associated with a 2.8-fold increased risk of fibrosis, indicating its potential use as a biomarker triad for post-COVID fibrotic risk stratification [50].

The levels of TGF-β in patients with PC19-PF are significantly elevated, contributing to the pathogenesis of fibrosis and long-term respiratory complications. TGF-β is a key cytokine involved in the fibrotic response, and its overexpression has been linked to persistent immunosuppression and lung remodeling following COVID-19 infection. A prospective longitudinal study of 82 COVID-19 ARDS survivors investigated the role of TGF-β1 in early fibrotic changes and long-term lung function impairment. Plasma, exhaled breath condensate (EBC), and PBMCs were collected at 6- and 24-months post-ICU discharge. Elevated TGF-β1 levels in plasma and EBC at 6 months were associated with a reduced gas exchange capacity (DmCO/VA), dyspnea, and increased lung attenuation on CT. In vitro, plasma and EBC from survivors induced profibrotic activation in primary human fibroblasts—marked by increased α-SMA and phospho-MARCKS expression—in a TGF-β-dependent manner. The study highlights the pivotal role of the TGF-b1 pathway in the early development of fibrotic abnormalities in COVID-19-induced ARDS survivors and demonstrates its predictive value for long-term functional impairment [65].

A comparative transcriptomic study using bulk RNA sequencing on small airway cell cultures from lung explants of IPF, PC19-PF, and control patients revealed distinct molecular signatures. While IPF cultures showed enriched interferon signaling, cell cycle regulation, and elevated bone morphogenetic protein (BMP) signaling (with nuclear pSMAD1/5/8 localization), PC19-PF cultures demonstrated enhanced TGF-β1 activity, ECM remodeling, macrophage classical signaling pathway activation, along with the acute phase response signaling pathway, and cytoplasmic pSMAD1/5/8 localization. Differential expression and pathway analyses indicated that PC19-PF is marked by persistent TGF-β1-driven fibrotic and immune pathways, whereas IPF showed elevated BMP signaling, which appears to inhibit TGF-β1 signaling [83].

### 3.2. Animal Studies

To better understand the pathogenesis of long COVID and PC19-PF and to facilitate drug development, suitable animal models are needed. A wide range of animal models, from mice to ferrets, hamsters, non-human primates, and cats, were used for studying SARS-CoV-2 infection but only a few models effectively replicate the chronic fibrotic lung damage seen in humans. Among those models, transgenic hACE2 mice and mouse-adapted SARS-CoV-2 models are most used due to their ability to replicate chronic lung fibrosis and immune dysregulation, particularly in aged mice [84,85,86,87,88]. Here, we focus on the animal models that could best recapitulate PC19-PF, offering critical insights into its progression and potential therapeutic targets (Table 2).

#### 3.2.1. Mouse Models

Due to their biological resemblance to humans, distinct immune systems, and rapid reproductive and growth rates, mice have proven to be a great model for investigating a variety of infectious diseases that affect humans. Using the mouse model for SARS-CoV-2 infection studies is challenging because of their resistance to the virus due to evolutionary variations in ACE2 receptors. To address these limitations, researchers have either adapted the SARS-CoV-2 virus to mice [87,92,93] or generated transgenic mice that express human ACE2 (hACE2) [94]. These approaches made mice susceptible to infection, making them effective models for exploring the pathogenesis, molecular mechanisms, and therapeutic targets for SARS-CoV-2 infection. These models have provided critical insights for understanding PC-19PF, as discussed below.

Li et al. researched to understand the mechanism of respiratory PASC (R-PASC) by comparing the BAL fluid and scRNA-seq data from clinical human PASC samples and relevant PASC mouse models. SARS-CoV-2 MA10-infected aged C57BL/6J mice (21 months old) were used to mimic the immune profile of R-PASC individuals, while SARS-CoV-2 MA10-infected aged BALB/c mice (around 1 year old) were used to study IFN-γ independence in another PASC model. BAL cells and lung samples were collected at 0, 3, 10, 21, and 35 days post-infection (dpi). The study identified a strong pro-fibrotic monocyte-derived macrophage (MoAM) response and abnormal interaction between pulmonary macrophages and respiratory resident T cells in both R-PASC human and mouse samples. Notably, IFN-γ was identified to cause chronic inflammation and fibrosis in mice, and blocking IFN-γ after the acute infection resolved these issues and improved lung function. This suggests that IFN-γ may have a crucial role in the development of R-PASC and is a potential target for future therapies [84].

Complementing these findings, Cui et al. investigated the mechanisms behind PC19-PF using a humanized mouse model and patient tissue analysis. They identified chronic immune activation in the lung tissues from long COVID patients, which was characterized by increased expression of CD47, IL-6, and phospho-JUN (pJUN), a key feature of this condition. To understand it better, the researchers developed a mouse model by intranasally co-transducing (6–12-week-old) mice with a human ACE2 lentivirus and a SARS-CoV-2 pseudovirus, with inducible JUN expression to recapitulate the lung fibrosis seen in patients. They also used a humanized mouse model carrying human lung organoids subcutaneously infected with the pseudo-virus to further study the human-specific aspects of the disease. The study highlights the IL-6-JUN-CD47 axis as a critical pathway in PC-19PF development and an abundance of inflammatory cells like matured neutrophils and MoAM in fibrotic lungs, contributing to IL-6 production and leading to the fibrotic pathway. The combined blockade of IL-6 and CD47 effectively ameliorated fibrosis in both mouse models and restored innate immune equilibrium, suggesting a potential therapeutic strategy for PC-19-PF [85].

Wu et al. investigated the lasting lung damage and the progression of pulmonary fibrosis caused by SARS-CoV-2 utilizing the humanized CD147 transgenic (hCD147) mouse model. The mice were infected intranasally with SARS-CoV-2, and major tissue samples were collected and analyzed at different time points: 2, 6, 13, 20, and 27 dpi. The study found that the CD147 receptor contributes to SARS-CoV-2-induced lung damage. Histopathological analysis showed notable fibrotic and inflammatory changes in the diseased lungs. RNA sequencing analysis showed a marked increase in profibrotic genes and other key proteases (MMP3, MMP10, and ADAMTS4), ECM proteins (e.g., collagens, ELN, MFAP4, and THBS4), and pro-fibrotic cytokines and chemokines (IL-4, IL-6, IL-17, IFN-γ, CCL2, CCL4, IL-10, CXCL9, and CXCL10). The study also demonstrated a significant increase in the principal profibrogenic cytokine TGF-β and related signaling pathways in the lungs of infected mice [86].

Dinnon et al. conducted a study infecting 1-year-old female BALB/c mice with the mouse-adapted SARS-CoV-2 strain MA10 to investigate the pathogenesis of lung abnormalities associated with PASC. Histological analysis of lung tissue collected from 15 to 120 days after viral clearance revealed subpleural lesions characterized by the presence of collagen deposits, proliferating fibroblasts, and chronic inflammation, including tertiary lymphoid structures. Longitudinal spatial transcriptional profiling uncovered the widespread dysregulation of reparative and fibrotic pathways within affected regions, mirroring the pathophysiological features observed in humans with COVID-19. Furthermore, populations of alveolar intermediate cells, along with the focal upregulation of pro-fibrotic markers, were identified in persistently diseased areas [87].

Giannakopoulos et al. utilized 8–12-week-old K18-hACE2 mice infected intranasally with SARS-CoV-2 (USA-HI-B.1.429 strain) to analyze the post-COVID pulmonary pathology. The lungs of the mice were evaluated at 5 and 30 dpi through pathological and immunological analyses. At 5 dpi, mice exhibited interstitial thickening, immune cell infiltration, widespread collagen deposition, and alveolar epithelial damage marked by reduced AT1 (Hop+) and AT2 (Pro-SPC+) cell markers. By 30 dpi, airway wall thickening worsened, with persistent inflammation, ectopic iBALT (Inducible Bronchus-Associated Lymphoid Tissue) formation, intense peribronchiolar collagen deposition, and a sustained loss of Sftpc expression. There were notable increases in CD45+ and Ly6G+ cells, elevated levels of cytokines (IL-6, IL-10, and IFNγ) and chemokines (RANTES, MCP-1, and CXCL10), and significant NET formation (CitH3 + MPO+), highlighting prolonged epithelial injury and unresolved inflammation contributing to post-viral lung sequelae [88].

In summary, these studies have provided mechanistic insights into SARS-CoV-2-induced pulmonary fibrosis, revealing critical roles for the IFN-γ, IL-6/JUN/CD47, and TGF-β pathways, as well as identifying promising therapeutic targets, such as cytokine blockade, for treating PC19-PF.

#### 3.2.2. Syrian Hamster Model

Syrian hamsters were among the first models used in COVID-19 research due to their ACE2 receptor’s similarity to humans [95]. Upon SARS-CoV-2 infection, they develop a non-lethal illness with weight loss and severe lung pathology resembling human COVID-19 [96]. In a longitudinal study by Li et al., they revealed persistent alveolar-bronchiolization, fibrosis, and viral RNA presence up to 120 days post-infection. These changes were associated with sustained Notch signaling and aberrant activation of CK14^+^ basal cells, indicating impaired repair and chronic inflammation [89]. Aged hamsters exhibited biphasic respiratory dysfunction, acute impairment followed by prolonged exercise intolerance, mirroring long COVID. Transcriptomics showed early immune activation transitioning into chronic ECM remodeling and vascular inflammation [90]. 

Sex-specific analyses revealed divergent recovery: males displayed prolonged proteomic disruption, while females recovered faster but showed persistent impairment in SCGB1A1, a lung-protective protein. Both sexes had elevated levels of mucus-related proteins, indicating ongoing respiratory stress [91].

Despite limited diagnostic and molecular tools compared to mice, Syrian hamsters remain a valuable model for post-COVID-19 research due to their ability to replicate key human PASC features, such as viral persistence, sex-specific responses, and aberrant lung repair. While most studies focus on short-term outcomes (<3 months post-infection), they consistently reveal persistent or progressive tissue injury originating during acute infection. These findings underscore the potential for ongoing pathological changes beyond the current observation windows and highlight the model’s relevance for future long-term mechanistic and therapeutic investigations.

### 3.3. Shared Mechanisms with Other Fibrotic Lung Diseases and Pharmacological Interventions

Interestingly, the abovementioned molecular insights into the pathogenesis of PC19-PF share similar mechanisms with other pulmonary conditions like fibrotic interstitial lung diseases (f-ILDs) or IPF. A study performed scRNA-seq on patients with pathological progression from subacute COVID-19 to the organizing diffuse alveolar damage (ODAD) phase and ultimately chronic PF. This transition had the upregulation of transcription factors SMAD2 and SMAD3, key regulators of the TGF-*β* signaling pathway, which is essential for driving the EMT and the progression of fibrotic diseases, including PC19-PF. The study also identified 86 genes commonly expressed in patients with PC19-PF and f-ILDs. Many of these genes were involved in WNT and cadherin signaling pathways, both of which drive the EMT and profibrotic cascade. Furthermore, higher expression of MUC5ac and WNT10a in patients with PC19-PF and f-ILDs highlights their roles in PF [97].

Another study used 33 autopsy lung samples from COVID-19 patients and identified a significant upregulation of fibrotic genes such as COL1A2, COL3A1, COL6A3, and LOX, which are associated with collagen synthesis, extracellular matrix (ECM) remodeling, and fibroblast activation. This pattern is similarly observed in IPF patients, suggesting shared mechanisms of fibrosis in both diseases. Furthermore, pathological fibroblasts (pFBs) expressing markers such as CTHRC1 and SPARC were found to be enriched in PC-19-PF and IPF patients, indicating a common fibrotic process and supporting the possibility of repurposing antifibrotic therapies approved for IPF in the treatment of PC19-PF [76].

Given the overlap between PC19-PF and other fibrotic lung diseases, especially IPF, several therapies approved for IPF are being considered for repurposing. Shared mechanisms include the upregulation of TGF-β/SMAD signaling, EMT activation, collagen deposition, and expansion of pathological fibroblasts, as seen in transcriptomic and autopsy studies [76,97]. Currently, only two antifibrotic medications, nintedanib and pirfenidone, have been approved by regulatory bodies such as the United States Food and Drug Administration (FDA) and the European Medicines Agency (EMA) for the treatment of PF [98,99] and are under investigation for treating PC19-PF. The clinical rationale for using antifibrotic therapy in COVID-19 patients includes preventing complications from ongoing infection, stimulating the recovery phase, and controlling the fibroproliferative processes

Pirfenidone, an antifibrotic agent, works by inhibiting the recruitment and accumulation of inflammatory cells, impeding fibroblast proliferation, reducing extracellular matrix deposition, and disrupting SARS-CoV-2 pathogenesis by inhibiting furin, a critical protein required for viral entry. Furthermore, it regulates signaling pathways associated with PC19-PF. These dual anti-inflammatory and antifibrotic properties suggest a potential therapeutic approach to slowing the progression of PC19-PF [100].

Nintedanib, an alternative therapeutic option, is a tyrosine kinase inhibitor that targets the fibroblast and myofibroblast cascades, potentially affecting pulmonary angiogenesis. Notably, clinical trials such as INPULSIS have demonstrated its capacity to attenuate the decline in forced vital capacity (FVC) observed in IPF patients, thus impeding disease progression over a brief period of 4–6 weeks [101]. Antifibrotic therapies show promise in slowing the progression of fibrosis in patients with confirmed radiological evidence of fibrosis and an increased risk of disease progression [25].

A few trials, such as FIBRO-COVID and NINTECOR, are currently evaluating the safety and efficacy of pirfenidone and nintedanib in the treatment of PC19-PF (ClinicalTrials.gov Identifiers NCT04607928 and NCT04541680). However, due to the distinct etiology, clinical course, and inflammatory profile of PC19-PF, there is still no definitive evidence supporting the effectiveness of antifibrotic therapies in this setting. Furthermore, side effects of antifibrotic drugs—including diarrhea, fatigue, and loss of appetite—overlap with COVID-19 symptoms, potentially hindering an early diagnosis and worsening clinical manifestations, which must be carefully managed in this patient population.

In terms of immunomodulation, dexamethasone is a widely used, low-cost steroid that significantly reduces mortality in hospitalized COVID-19 patients who require oxygen or ventilatory support. It works by suppressing immune-mediated inflammation that causes lung injury and organ failure in severe cases. Its benefits are most pronounced after the first week of illness, when immune dysregulation outweighs viral replication. However, glucocorticoid therapy might have rare but some serious side effects, including high blood sugar levels, gastrointestinal bleeding, and psychosis. In the context of PC19-PF, the role of corticosteroids is less clearly defined and may be limited to patients with active inflammation or organizing pneumonia features on imaging or histology [102].

Among targeted cytokine therapies, ruxolitinib, a JAK1/2 inhibitor, reduces cytokine-driven hyperinflammation in COVID-19 patients by blocking the JAK/STAT pathway. In the RuxCoFlam trial, it led to a ≥25% reduction in inflammation scores in 71% of patients, lowered the levels of cytokines (IL-6, TNF-α, and IFN-γ), and was well tolerated. However, its benefits are limited by a lack of randomized data, confounding from co-treatments, and reduced efficacy in early or very late-stage disease. Its use is most effective in patients with clearly defined hyperinflammation. This suggests that while ruxolitinib might help prevent progression in the inflammatory phase of COVID-19, its direct role in reversing established fibrosis, such as in PC19-PF, remains uncertain [103].

Similarly, IL-6 receptor antagonists such as tocilizumab and sarilumab have shown a benefit in hospitalized patients with systemic hyperinflammation, particularly when used in combination with corticosteroids. These agents reduce the exaggerated IL-6-mediated immune response implicated in lung injury and ARDS. While they have been shown to reduce mortality and the need for mechanical ventilation in severe COVID-19 patients, their role in post-viral fibrotic remodeling is not well-established. Moreover, risks of secondary infections and liver enzyme elevation necessitate cautious use [104].

In addition, other molecules with antifibrotic properties could be considered in treating PC19-PF. Pamrevlumab, also known as FG-3019, is a humanized monoclonal antibody that targets and inhibits connective tissue growth factor (CTGF). CTGF is produced by damaged epithelial alveolar and endothelial cells and promotes vascular leakage, chemotaxis, and fibrotic changes, all of which contribute to the pathogenesis of fibrotic lung disease [105,106]. 

The IN01 vaccine is a recombinant molecule that induces the immune system to generate polyclonal anti-epidermal growth factor (EGF) neutralizing antibodies. The vaccine’s goal in eliciting these antibodies is to prevent the activation of the epidermal growth factor receptor (EGFR), which in turn inhibits downstream signaling pathways. This EGFR signaling inhibitor has the potential to reduce the hyperimmune response to lung injury seen during SARS-CoV-2 infection [107,108].

In summary, while several therapeutic options shows promises for PC19-PF based on the mechanistic rationale and extrapolation from IPF or severe COVID-19 cohorts, the evidence base remains limited. The heterogeneous nature of PC19-PF, with some patients demonstrating spontaneous improvement and others progressing to irreversible fibrosis, underscores the need for precise patient stratification, timing of the intervention, and personalized therapeutic approaches. Until robust, prospective, and controlled trials confirm the safety and efficacy of these agents in PC19-PF, their use should be considered experimental.

## 4. Prevention Strategies to Mitigate PC19-PF

Given the lack of definitive treatments for PC19-PF, prevention is crucial in reducing the risk of PC19-PF. Since severe COVID-19 is the main driver of fibrotic lung damage, minimizing disease severity is key. Vaccination remains the most effective tool, significantly lowering mortality and hospitalization across all major variant waves (Pre-Alpha to Omicron), as shown in a large South Carolina cohort study of over 860,000 adults [109]. By reducing severe illness, vaccination helps prevent lung injury and fibrotic sequelae.

Additional preventive strategies such as mask-wearing, hand hygiene, physical distancing, and improved ventilation in diagnosed individuals remain important for curbing viral spread and associated complications. Emerging evidence also shows that newer SARS-CoV-2 variants like Omicron cause milder lung disease, especially in vaccinated populations. Booster doses further reduce pneumonia severity [110,111]. Moreover, early use of antivirals like nirmatrelvir/ritonavir has been linked to a reduced risk of post-COVID lung complications, regardless of prior vaccination or infection, likely by limiting viral replication and inflammation [112].

Preventive efforts should also extend into the acute and post-acute phases of illness. During acute infection, corticosteroids, antivirals, and lung-protective ventilation strategies can reduce lung damage. Following recovery, early pulmonary rehabilitation, regular lung function and imaging follow-up, and healthy lifestyle changes support lung healing. Patient education, adherence to care, and accessible follow-up are also key components of long-term prevention and recovery.

## 5. Conclusions

In summary, PC19-PF is a severe and persistent consequence of SARS-CoV-2 infection that has significant effects on both infected individuals and global healthcare systems. This review has addressed the clinical presentation and underlying pathophysiology, highlighting the complex nature of long-term COVID-induced lung fibrosis. To advance the field, several critical research priorities must be addressed. These include identifying early diagnostic and prognostic biomarkers for fibrosis progression, understanding the temporal evolution and potential reversibility of fibrotic lesions, defining patient subgroups most likely to benefit from antifibrotic or immunomodulatory therapy, and evaluating the long-term safety and efficacy of both repurposed and novel therapeutics in well-designed clinical trials. In addition, future studies should aim to characterize the psychosocial and extrapulmonary sequelae of PC19-PF, which remain underrecognized despite their clinical relevance. Addressing these gaps will require a multidisciplinary approach that integrates basic science, clinical observations, and translational research to develop evidence-based, patient-centered strategies for the early detection, monitoring, and treatment of PC19-PF.

## Figures and Tables

**Figure 1 viruses-17-01098-f001:**
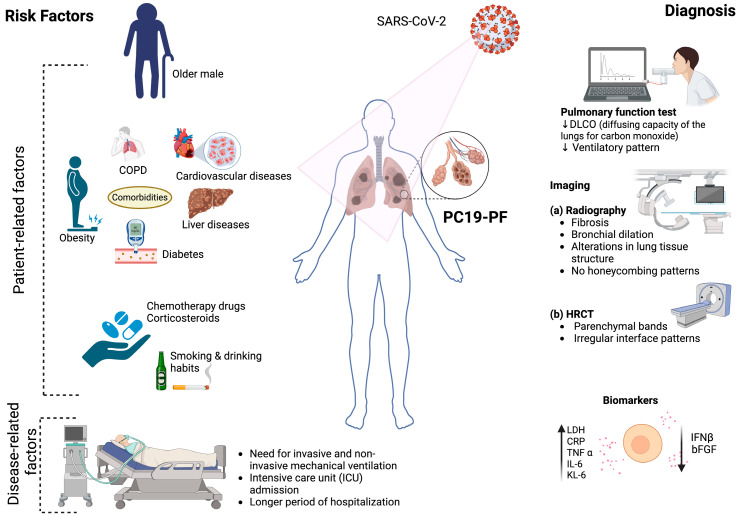
PC19-PF-associated risk factors and their diagnostic approaches. (Created with biorender.com).

**Figure 2 viruses-17-01098-f002:**
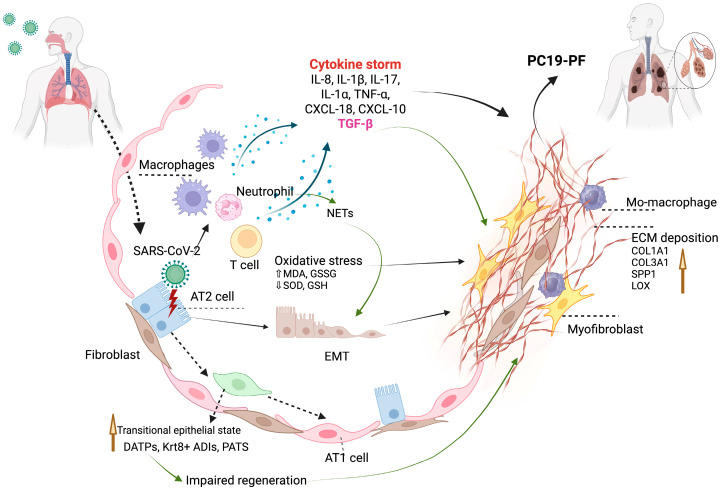
Illustration of the potential mechanisms of PC19-PF (created with biorender.com).

**Table 1 viruses-17-01098-t001:** Biomarkers potentially associated with PC19-PF.

Biomarkers	Higher Concentration	Lower Concentration	References
Acute phase proteins	CRP, C5b-9	albumin	[50,51,52]
Biochemical markers	LDH ALT, AST		[13,50,53,54]
Cytokines and chemokines	CXCL-10, CCL-18, TNF α, TGF β, IL-1 α, IL-6	INF-α, INF β, IFN γ	[28,50,55,56]
Vascular markers	VCAM-1, ICAM-1, PIGF	bFGF	[57,58]
Mitochondrial regulatory proteins	PINK1, DNM1L, MFN2		[56]
Others	α-SMA, COL1A2, COL3A1		[59]
	KL-6		[57]
	MMP1, MMP7		[60]
	SARS-CoV-2 Ig G		[61]

CRP, C-reactive protein; C5b-9, terminal complement complex; LDH, lactate dehydrogenase; AST, aspartate aminotransferase; ALT, alanine transaminase; CXCL10, C-X-C motif chemokine ligand 10; CCL-18, chemokine (C-C motif) ligand 18; TNFα, tumor necrosis factor alpha; TGF β, transforming growth factor β; IL-1α, interleukin 1α; IL-6, interleukin 6; IFN-α, interferon alpha; IFN-β, interferon beta; IFN-γ, interferon gamma; VCAM-1, vascular cell adhesion molecule-1; ICAM-1, intercellular adhesion molecule-1; PIGF, placental growth factor; bFGF, basic fibroblast growth factor; PINK1, PTEN-induced kinase 1; DNM1L, dynamin-1-like protein; MFN2, mitofusin-2; α-SMA, alpha-smooth muscle actin; COL1A2, collagen type 1 alpha 2, COL3A1, collagen type 3 alpha 1; KL-6, Krebs von den Lungen 6, MMP1 and MMP7, matrix metalloproteinases 1 and 7.

**Table 2 viruses-17-01098-t002:** SARS-CoV-2 animal models used to study respiratory outcomes of long COVID.

Authors	Species	Age	Sex	Virus Used	Duration of Study	Important Pulmonary Lesions/Findings	Ref.
Li et al.	C57BL/6J and BALB/c mice	21 mons and 1 yr	Female	SARS-CoV-2 MA10 intranasal	0–35 dpi	Pro-fibrotic MoAM response; abnormal interaction of MoAM and T cells; IFN-γ-driven fibrosis; and blocking it reduces the pathology	[84]
Cui et al.	lsl-rtTA JUN (B6/129) mice	6–12 wks	Male and female	SARS-CoV-2 pseudovirus and hACE2 lentivirus (intranasal and aerosol)	27 dpi	Fibrosis driven by the IL-6-JUN-CD47 axis; increased MoAMs and neutrophils; fibrosis reversed by IL-6 and CD47 blockade	[85]
	Humanized Nod.Scid.IL2RG-/-NSG mice	4 mon	Not stated	SARS-CoV-2 pseudovirus subcutaneous (lung graft)	48 dpi	Human-like lung fibrosis in engrafted lung tissue; fibrosis reduced by CD47/IL-6 inhibition	
Wu et al.	hCD147 transgenic mice	Not stated	Not stated	SARS-CoV-2 (wild-type and Delta variant) intranasal	2, 6, 13, 20 and 27 dpi	CD147-driven fibrosis; ECM and collagen deposition; pro-fibrotic cytokine surge; fibrosis reversed by meplazumab	[86]
Dinnon et al.	BALB/c and C57BL/6J mice	10 wks and 1 yr	Female	SARS-CoV-2 MA10 intranasal	2 to 120 dpi	1 year old Balb/c mice mimic fibrotic changes after SARS-CoV2 MA10 infection; chronic pulmonary lesions; subpleural fibrosis; tertiary lymphoid structures; antifibrotic and antiviral interventions are effective	[87]
Giannakopoulos et al.	K18-hACE2 mice	Not stated	Not stated	SARS-CoV-2 (USA-HI-B.1.429, Delta-like) intranasal	5 and 30 dpi	Persistent lung inflammation; NETs; epithelial injury; collagen deposition; prolonged cytokine and neutrophil infiltration; iBALT formation and thickened airway walls at 30 dpi	[88]
Li et al.	Golden Syrian Hamster	6–8 weeks old	Male	SARS-CoV-2 wild-type strain HK-13 and BA.5 strains (intranasal)	Up to 120 dpi	Chronic inflammation and multifocal alveolar bronchiolization; CK14^+^ basal cells proliferate into club and ciliated cells; Notch3 pathway activation; viral RNA/protein persistence in macrophages	[89]
Heydemann et al.	Golden Syrian Hamster	1 yr	Male	SARS-CoV-2 Delta variant intranasal	0–112 dpi	Persistent lung fibrosis, bronchiolization, CK14^+^/SCGB1A1^+^ cell proliferation; impaired exercise recovery; persistent viral RNA (RdRp); ECM remodeling; chronic transcriptomic changes; vascular remodeling gene upregulation; no chronic lesions in other organs; the model mimics respiratory long-COVID	[90]
Boese et al.	Golden Syrian Hamster	6 wks	Male and Female	SARS-CoV-2 (ancestral strain)	1, 3, 5 and 31 dpi	Sex-based lung proteomic changes; persistent mucus-related protein elevation (Mucin 5B, CLCA-1) at 31 dpi; male-specific persistent downregulation of proteins; overlapping pathways (coagulation and complement cascades); delayed return to baseline in males; SCGB1A1 altered in females; neurodegenerative pathway signatures in both sexes at 5 dpi	[91]

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
