# Peer review of "Breathless Aftermath: Post-COVID-19 Pulmonary Fibrosis"

_viruses, 2025, doi:10.3390/v17081098_

Round 1
Reviewer 1 Report
Comments and Suggestions for Authors
The article covers a broad spectrum of relevant questions related to PC19-PF, both from the clinical and experimental perspectives.
The paper effectively integrates current knowledge on the topic and presents it in an orderly manner.
The introduction discusses the effect of COVID-19 but recommend emphasizing PC19-PF as a unique condition, different from other fibrotic lung diseases, that may demarcate the article's value to this field.
Prevalence from various studies are stated in the text, but these would be more usefully summarized with some explanation of the reasons for variability of the findings. In addition, more emphasis on the biological mechanisms driving the risk factors would strengthen this part of the analysis.
Translating the terms and concentrating on the primary mechanisms of Post-COVID-19 Pulmonary Fibrosis may help the section become more readable for a larger audience.
The section on animal models is comprehensive but somewhat redundant. The Syrian hamster model section, in particular, can be condensed since it does not add much that is different from the utilization of mouse models. Moreover, the common mechanisms with other fibrotic lung diseases can more directly relate to clinical findings in humans, with a focus on how such common mechanisms can be translated into therapeutic strategies for PC19-PF.
Although the therapeutics chapter covers antifibrotic medications such as nintedanib and pirfenidone, the review would benefit from a more complete consideration of how these may be used in PC19-PF and the pitfalls of extrapolating these therapies from IPF to PC19-PF.The coverage of corticosteroids, anti-JAKs, and anti-IL-6 therapy is also brief and would be improved by more on their efficacy in long COVID.
The conclusion could provide specific calls to action for researchers and clinicians. While recommending further studies, it would be useful to highlight the most important research questions or knowledge gaps that need to be addressed.

Author Response
Dear Editor,
We thank you for the opportunity to revise the manuscript. We also thank the reviewer for the constructive and thoughtful comments for improving this manuscript. We have revised our manuscript based on the reviewer’s comments. The changes have been highlighted in red fonts. The following are the point-to-point responses to the reviewers’ comments.
Reviewer 1:
Comment 1: The introduction discusses the effect of COVID-19 but recommend emphasizing PC19-PF as a unique condition, different from other fibrotic lung diseases, that may demarcate the article's value to this field.
Response: We have revised the introduction to highlight post-COVID-19 pulmonary fibrosis (PC19-PF) as a distinct condition, separate from other fibrotic lung diseases such as idiopathic pulmonary fibrosis (IPF) or ILDs. We now emphasize its unique post-viral origin, variable course, and potential for reversibility in some patients. This distinction helps clarify the focus and relevance of our review within the broader context of fibrotic lung diseases.
We have added the following sentence to the introduction (paragraph 3): Unlike idiopathic pulmonary fibrosis (IPF) or other interstitial lung diseases (ILDs), PC19-PF arises as a sequela of viral pneumonia and often shows a different clinical pattern, including distinct imaging features and, in some cases, partial recovery over time (Line 44 to 47).
Comment 2: Prevalence from various studies is stated in the text, but these would be more usefully summarized with some explanation of the reasons for variability of the findings. In addition, more emphasis on the biological mechanisms driving risk factors would strengthen this part of the analysis
Response: We have revised Sections 2.1 and 2.2 to improve clarity and depth. Specifically:
Section 2.1 (Epidemiology) has been revised to more clearly summarize prevalence findings from major studies and meta-analyses. Specifically, we now discuss the heterogeneity in study design (e.g., cohort vs. cross-sectional), inclusion criteria (e.g., hospitalized vs. outpatient populations), timing of post-COVID assessment, sample sizes, outcome definitions (clinical vs. radiological fibrosis), and data collection methods in each study quoted. We believe these additions provide important context for understanding the wide range of prevalence estimates and improve the interpretability of the reported data.
Changes are made in pages 2 and 3, lines 58-60, 63-72, 76-80 and 84-96
Section 2.2 (Risk Factors): We have strengthened this section by explicitly highlighting the biological mechanisms underlying key risk factors for PC19-PF. While our original text discussed immune dysregulation, epithelial injury, and impaired tissue repair in relation to some risk factors, we have now expanded these discussions and integrated them more clearly throughout the section. Specifically, we now detail how aging-related mechanisms (such as immunosenescence, telomere attrition, and ECM remodeling), sex-based immune differences, and comorbidities (including hypertension, diabetes, and COPD) contribute to dysregulated repair and fibrosis after SARS-CoV-2 infection. We have also added mechanistic insights on how lifestyle factors (e.g., smoking, alcohol) and the use of immunosuppressive therapies can promote a pro-fibrotic state.
These additions aim to provide a more comprehensive understanding of how various risk factors biologically predispose individuals to PC19-PF and account for the heterogeneity observed across clinical cohorts.
All changes have been incorporated in the revised manuscript on page [3-5], section 2.2. The section has also been reorganized in alignment with Reviewer 2’s suggestions.
Comment 3: Translating the terms and concentrating on the primary mechanisms of post-COVID-19 Pulmonary Fibrosis may help the section become more readable for a larger audience.
Response: We have revised the section to simplify and clarify the descriptions of the primary mechanisms underlying PC19-PF. Technical terms have been explained where necessary, and the mechanistic overview has been rewritten in a more accessible format to ensure better readability for a broader audience.
All changes have been incorporated in the revised manuscript on page no:10-12, lines 403-411, 426-434, 471, 477-479.
Comment 4: The section on animal models is comprehensive but somewhat redundant. The Syrian hamster model section, in particular, can be condensed since it does not add much that is different from the utilization of mouse models.
Response: We have condensed the Syrian hamster model section to reduce redundancy with mouse model descriptions, focusing on key unique contributions such as viral persistence, biphasic respiratory dysfunction, and sex-specific recovery differences. The mouse model section remains comprehensive given the mechanistic diversity and therapeutic insights provided by different mouse studies. This revision improves clarity and flow while preserving all critical information and adhering to your suggestions.
All changes have been incorporated in the revised manuscript on page no: 18, under 3.2.2. Syrian hamster model.
Comment 5: Moreover, the common mechanisms with other fibrotic lung diseases can more directly relate to clinical findings in humans, with a focus on how such common mechanisms can be translated into therapeutic strategies for PC19PF.
Although the therapeutics chapter covers antifibrotic medications such as nintedanib and pirfenidone, the review would benefit from a more complete consideration of how these may be used in PC19-PF and the pitfalls of extrapolating these therapies from IPF to PC19-PF.
The coverage of corticosteroids, anti-JAKs, and anti-IL-6 therapy is also brief and would be improved by more on their efficacy in long COVID.
Response: We have expanded the therapeutic section to more clearly delineate the shared molecular mechanisms between PC19-PF and other fibrotic lung diseases such as IPF, including TGF-β signaling, EMT, and fibroblast activation. These mechanistic overlaps provide a clinical rationale for repurposing antifibrotics like nintedanib and pirfenidone in PC19-PF. However, we have also discussed the limitations of directly extrapolating treatment efficacy from IPF to PC19-PF, given differences in disease etiology, reversibility, and immune context.
Additionally, we have revised and extended the sections on corticosteroids, JAK inhibitors, and anti-IL-6 therapies and target molecules to better address their roles and emerging evidence in PC19-PF management, particularly in mitigating inflammation-driven fibrotic progression. These changes aim to improve the translational value and clinical relevance of the therapeutic discussion.
All changes have been incorporated in the revised manuscript on page no: 18 to 20, under 3.3.
Comment 6: The conclusion could provide specific calls to action for researchers and clinicians. While recommending further studies, it would be useful to highlight the most important research questions or knowledge gaps that need to be addressed.
Response: We have revised the conclusion to explicitly identify critical knowledge gaps and research priorities. The updated section now highlights the need for mechanistic studies, improved diagnostic tools, and targeted therapeutic strategies, with an emphasis on longitudinal and translational research approaches. We have also included specific recommendations for future clinical and research efforts to better inform patient care and guide treatment development. All the changes are incorporated in the revised manuscript on page no: 20-21 under conclusion part.
Reviewer 2:
General Comment: I generally think that the authors must focus on FIBROSIS and less on long COVID. As I wrote in my comments below, fibrosis means fibrotic changes – hence, addressing fibrosis as "symptoms" (without imaging) or with only GGO present is not really assessing fibrosis.
Response: We agree that fibrosis should be defined based on radiological or histopathological evidence of fibrotic changes and not solely inferred from persistent symptoms or ground-glass opacities (GGOs), which may reflect ongoing inflammation rather than true fibrosis. In response, we have revised the manuscript to maintain a clear focus on post-COVID fibrotic lung disease specifically, and we have removed or reworded instances where fibrosis may have been ambiguously referenced. We now emphasize imaging-confirmed fibrotic features—such as reticulations, traction bronchiectasis, and architectural distortion—as more reliable indicators of PC19-PF. These revisions help distinguish fibrosis from broader “long COVID” manifestations and ensure our review remains centered on the fibrotic sequelae of SARS-CoV-2 infection.
Changes have been incorporated throughout the text, particularly in the sections discussing prevalence, clinical features, and diagnosis.
Some specific comments:
Comment 1: Epidemiology – you should emphasis in each of the examples of prevalence that you give if this is out of patients with severe covid / hospitalized/ general population with COVID. Obviously, as most patients have mild COVID, less than 10% have fibrosis and this must be highlighted.
Response: We have clarified the clinical context of each prevalence estimate by indicating whether the data were derived from hospitalized, severe/critical, or mixed COVID-19 populations. We have also emphasized that the high prevalence rates reported in meta-analyses largely reflect moderate-to-severe or hospitalized cases and are not representative of the general COVID-19 population. To address this further, we have added data highlighting that only approximately 9% of patients with severe or critical disease are estimated to develop fibrotic-like changes, and that in the general population of COVID-19 survivors, the overall prevalence remains significantly lower. These revisions aim to ensure a clear and accurate interpretation of the epidemiological burden of post-COVID fibrosis.
All changes made are included in page 2 under the epidemiology section.
Comment 2: You should provide, at least for your major examples in each part, which variant/s were assessed that the findings refer to. Nowadays, with the Omicron variant, there are completely different characteristics to the infection, making many of your examples irrelevant. Therefore, you should also include more updated examples addressing the Omicron variant.
Response: We have incorporated a retrospective observational study conducted at Cleveland Clinic (March 2021–March 2022), which directly compares patients infected with Alpha, Gamma, Delta, and Omicron variants (n = 2,779). This study showed markedly milder clinical outcomes in Omicron-infected patients (e.g., 5.9% hospitalization, 1% ICU admission) in contrast to earlier variants. We also included a reference indicating that despite Omicron’s higher rate of breakthrough infections, severe disease—and by extension, fibrotic sequelae—was more strongly associated with host factors (age, comorbidities) than viral strain alone.
Changes are included in page 3-4 under risk factor section: lines 119 to 131.
Additionally, we acknowledge that not all studies explicitly reported the variant type, especially those conducted early in the pandemic. To improve interpretability, we have now specified the study period (e.g., first wave) where variant data were unavailable. This contextual detail provides clarity regarding the likely circulating strain and helps readers assess the relevance of the findings in the current epidemiological context.
Comment 3: The most important risk factor is the severity of the disease. All other things are basically derivative factors out of it. You should therefore start this section by addressing this factor and add more data on it.
Response: We have now restructured the section to begin with disease severity as the primary risk factor for PC19-PF. We have emphasized its central role by explaining how the extent of acute lung injury, especially in moderate-to-severe cases, directly correlates with the risk of fibrotic remodeling. We believe these updates help establish a more logical flow and reinforce the idea that other risk factors, such as mechanical ventilation, ICU admission, comorbidities, and age, are often closely linked to disease severity.
The entire risk factor section 2.2 on page 3 to 5 has been reorganized as suggested.
Comment 4: "utilization of self- 64 reported symptoms versus medical record data collection", persisting symptoms of lung fibrosis" – what do you mean by that? Fibrosis is only diagnosed based on CT findings and how do you know the symptoms were from it?
Response: We agree that the diagnosis of pulmonary fibrosis should rely on objective radiological evidence, typically via high-resolution computed tomography (HRCT), and/or pulmonary function testing. Fibrosis cannot be definitively diagnosed based on symptoms alone.
To clarify, our original mention of "self-reported symptoms" referred to methodological variability across studies, where some cohorts included patients reporting persistent respiratory symptoms post-COVID (e.g., dyspnea, cough) as part of long COVID surveillance, though not necessarily confirmed by imaging. In the revised draft, we have emphasized that studies included in the meta-analyses referenced here (particularly references 7 and 8) required imaging-based or clinical definitions that distinguished fibrotic from non-fibrotic cases. We have removed any language that could imply fibrosis was diagnosed based solely on symptoms.
Comment 5: The pulmonary functions are key in follow-up and identifying patients with possible COVID-fibrosis. The authors should elaborate on their progression over time. A prospective large study evaluated patients 3 and 6 months after the acute infection and while many had low DLCOc and other functions, there was no change in 3 months, and it did not correlate with improvements in CT scan in this period (DOI: 10.1016/j.rmed.2023.107367). The authors should use this unique data and others in this section.
Response: We have incorporated findings from a large prospective cohort study evaluated patients at 3 and 6 months post-infection. This study observed persistent impairments in DLCO and other lung function parameters without significant improvement during this interval, and notably, the changes in pulmonary function did not correlate with radiographic improvements. These findings underscore the complexity of PC19-PF progression and highlight the need for individualized and longitudinal pulmonary function monitoring beyond imaging alone. We have integrated this data and related studies into the pulmonary function section to provide a more comprehensive overview of functional progression in PC19-PF.
Changes are included in page 5-6 under pulmonary function test section: lines 206 to 227.
Comment 6: In the laboratory predictors, the authors should clearly mention for each variable if the levels found in correlation with fibrosis are from acute disease or those taken at follow-up when patients are diagnosed with fibrosis? In addition, once again evidence from omicron should be highlighted.
Response: We have updated the manuscript to clearly specify when each biomarker was measured in relation to the disease stage. All studies included in the biomarker section are follow-up studies on post-COVID-19 patients, specifically focusing on those with fibrosis. We have also clearly indicated where comparisons were made between patients with and without fibrosis.
Regarding SARS-CoV-2 variants, many studies did not specify the variant involved; therefore, we have noted the timing of the study or referenced the “first wave” where applicable to help contextualize the findings, considering the evolving nature of the pandemic and its variants.
Major changes related to the biomarkers related to pulmonary fibrosis are addressed in page 7-9.
Comment 7: Another laboratory variable that was shown to correlate with fibrosis in CT are levels of SARS-COV-2 antibodies. More specifically, anti-S levels were previously shown to be associated with changes in CT and lower pulmonary functions at follow-up.
Response: We have added SARS-CoV-2 IgG to Other biomarkers section (page 9, line 388-397)
Comment 8: The authors should include the importance of vaccinations, preventive measures and other proven interventions for prevention of severe COVID as a main tool to avoid fibrosis given the major role of severe disease.
Response: We have a new section “Prevention Strategies to Mitigate PC19-PF” (page 20). We emphasized vaccination, early antiviral use, and supportive care as key measures to reduce the risk of PC19-PF. These interventions help prevent severe COVID-19, minimize lung injury, and support recovery, thereby lowering the likelihood of long-term fibrotic complications.

Reviewer 2 Report
Comments and Suggestions for Authors
Thank you for the opportunity to review this paper. The authors provide a review on pulmonary fibrosis after acute COVID-19. They address its presentation, prevalence, predictors, and possible treatments. Overall, the study topic is extremely interesting and relevant, although not that novel. The manuscript is well written and the authors address the main aspects of this disease.
I generally think that the authors must focus on FIBROSIS and less on long COVID. As I wrote in my comments below, fibrosis means fibrotic changes – hence, addressing fibrosis as "symptoms" (without imaging) or with only GGO present is not really assessing fibrosis. Some specific comments:
- Epidemiology – you should emphasis in each of the examples of prevalence that you give if this is out of patients with severe covid / hospitalized/ general population with COVID. Obviously, as most patients have mild COVID, less than 10% have fibrosis and this must be highlighted.
- You should provide, at least for your major examples in each part, which variant/s were assessed that the findings refer to. Nowadays, with the Omicron variant, there are completely different characteristics to the infection, making many of you examples irrelevant. Therefore, you should also include more updated examples addressing the Omicron variant.
- The most important risk factor is the severity of the disease. All other things are basically derivative factors out of it. You should therefore start this section by addressing this factor and add more data on it.
- "utilization of self- 64 reported symptoms versus medical record data collection", persisting symptoms of lung fibrosis" – what do you mean by that? Fibrosis is only diagnosed based on CT findings and how do you know the symptoms were from it?
- The pulmonary functions are key in follow-up and identifying patients with possible COVID-fibrosis. The authors should elaborate on their progression over time. A prospective large study evaluated patients at 3 and 6 months after the acute infection and while many had low DLCOc and other functions, there was no change in 3 months and it did not correlate with improvements in CT scan in this period (DOI: 10.1016/j.rmed.2023.107367). The authors should use this unique data and others in this section.
- In the laboratory predictors, the authors should clearly mention for each variable if the levels found in correlation with fibrosis are from the acute disease or those taken at follow-up when patients are diagnosed with fibrosis? In addition, once again evidence from omicron should be highlighted.
- Another laboratory variable that were shown to correlate with fibrosis in CT are levels of SARS-COV-2 antibodies. More specifically, anti-S levels were previously shown to be associated with changes in CT and lower pulmonary functions at follow up.
- The authors should include the importance of vaccinations, preventive measures and other proven interventions for prevention of severe COVID as a main tool to avoid fibrosis given the major role of severe disease.
Author Response

(The authors gave the same response as above.)

Round 2
Reviewer 2 Report
Comments and Suggestions for Authors
Thank you for the opportunity to review this paper once again. The authors have addressed all my comments and the manuscript has significantly improved. I don't have additional comments that require a revision. Thank you and good luck.